# Genetic connectivity of the scalloped hammerhead shark *Sphyrna lewini* across Indonesia and the Western Indian Ocean

Sutanto Hadi[1], Noviar Andayani[2,3], Efin Muttaqin[3], Benaya M. Simeon[3], Muhammad Ichsan[3], Beginer Subhan[1], Hawis Madduppa[1]*

1 Department of Marine Science and Technology, Faculty of Fisheries and Marine Sciences, IPB University, Bogor, Indonesia, 2 Department of Biology, Faculty of Mathematics and Natural Science, University of Indonesia, Depok, Indonesia, 3 Wildlife Conservation Society Indonesia Program, Bogor, Indonesia

* hawis@apps.ipb.ac.id

## Abstract

Scalloped Hammerhead shark (*Sphyrna lewini*) is an endangered species which its populations have been declining globally including in Indonesia, the world's top shark fishing country. However, there is a lack of information on the recent population structure of this species to promote proper management and its conservation status. This study aimed to investigate the genetic diversity, population structure, and connectivity of the *S. lewini* population, in three major shark landing sites: Aceh (n = 41), Balikpapan (n = 30), and Lombok (n = 29). Meanwhile, additional sequences were retrieved from West Papua (n = 14) and the Western Indian Ocean (n = 65) populations. From the analyses of the mitochondrial *CO1* gene, a total of 179 sequences of *S. lewini*, with an average size of 594 bp, and 40 polymorphic loci in four and eight haplotypes for the Indonesian population and the Western Indian Ocean population were identified. The overall values of genetic diversity were high (h = 0.717; π = 0.013), with the highest values recorded in Aceh (h = 0.668; π = 0.002) and the lowest in Papua (h = 0.143; π = 0.000). On the contrary, the overall value was fairly low in the Western Indian Ocean (h = 0.232; π = 0.001). Furthermore, AMOVA and $F_{ST}$ showed three significant subdivisions in Indonesia ($F_{ST}$ = 0.442; $P$ < 0.001), with separated populations for Aceh and West Papua, and mixed between Balikpapan and Lombok ($F_{ST}$ = 0.044; $P$ = 0.091). In contrast, genetic homogeneity was observed within the population of the Western Indian Ocean ($F_{ST}$ = –0.013; $P$ = 0.612). The establishment of a haplotype network provided evidence of a significantly different population and a limited genetic distribution between the Indonesian and the Western Indian Ocean populations ($F_{ST}$ = 0.740; P < 0.001). This study showed the presence of a complex population of *S. lewini* with limited connectivity only in Indonesia separated from the Western Indian Ocean and requiring specific management measures based on the population structure at the regional level.

**Data Availability Statement:** All relevant data are within the paper. All sequences of S. lewini obtained were deposited in the BOLD System with the Barcode Index Number (BIN) registry of BOLD:

AAA2403 and database of GenBank with accession numbers MT324149-248.

**Funding:** This research was supported by Wildlife Conservation Society (WCS)-Indonesia, in collaboration with Faculty of Fisheries and Marine Sciences IPB University. SH received a scholarship from Indonesia Endowment Fund for Education (LPDP). The funders had no role in study design, data collection and analysis, decision to publish, or preparation of the manuscript.

**Competing interests:** The authors have declared that no competing interests exist.

## Introduction

The scalloped hammerhead shark, *Sphyrna lewini* is considered a coastal species because of its need for nursery areas. Globally, it is distributed in tropical waters as well as on the mainland, islands, and near the coastal region [1, 2]. This species has the unique modification of a lateral head which improves the ability to navigate and follow geomagnetic orientations across the ocean [3–5]. *S. lewini* can move in high rates of dispersal, and its female show allegiance to single nursery areas and exhibit no evidence of continued inter-oceanic migration. On the contrary, male is spread over a large area across ocean waters, with clear evidence of cross-reproduction and gamete transmission [6].

Scalloped hammerhead is one of the most exploited and threatened sharks. Around one to three million sharks are killed each year because of fishing and the shark fin trade around the world [7, 8]. This species was considered to be underexploited in 1999. However, in 2009 the International Union for Conservation of Nature (IUCN) listed the species on the Red List with Endangered (EN) status [9]. Five years later, the Convention on International Trade in Endangered Species (CITES) listed hammerhead sharks in Appendix II, and in 2019 the status was upgraded to Critically Endangered (CR) [10].

High exploitation of the *S. lewini* has an impact on its population structure, reducing the fecundity of the species and the genetic diversity [11]. Hammerhead sharks are viviparous with a yolk-sac placenta with an annual number of 12–30 young per litter. This species has a slow growth rate, late sexual maturity, long gestation period, and a long lifespan in nature [12–15]. The combination of high pressure and their biological properties makes this species vulnerable to overexploitation.

The study of population genetics has become an important tool for understanding population connectivity, supporting fisheries management, and improving conservation strategies. Furthermore, genetic information can be used to define the conservation effort and course of action by studying the structure of shark populations [16–19].

Shark fin product, including from scalloped hammerhead are very popular in Hong Kong [20], where trade regulations for endangered species and effective regulations are promoted [16, 20, 21]. The population structure of *S. lewini*, which is important for fisheries stock management, has been widely investigated in different coastal areas and ocean basins on a global and regional scale [17–19, 22, 23]. Duncan *et al.* [22] reported a global phylogeographic study of *S. lewini* that indicated that the Indo-West Pacific region is the center of diversity for tropical sharks, such as *S. lewini*, with a high and unique genetic diversity; however, no samples from Indonesia were included in that study, or in the study reported by Ovenden *et al.* [23], which included limited samples from Indonesia.

This study aimed to investigate the genetic diversity, population structure, and connectivity of *S. lewini*, where the populations of this species are affected by fishing activities at a regional scale in the Western Indian Ocean. Finally, the implications of these results for species management and conservation were examined.

## Material and methods

### Ethics statement

All samples were already dead when collected, and therefore, no approval from any institutional animal ethics committee was required. The sample collection and transportation followed the regulation of the Ministry of Marine Affairs and Fisheries of the Republic of Indonesia (Number 5/PERMEN-KP/2018) on the prohibition of cowboy and hammerhead shark export from Indonesia. Furthermore, this study was approved by the Ministry of Marine

Affairs and Fisheries of the Republic of Indonesia, under permission numbers 276/BPSPL.03/PRL/X/2018 and 319/PNK/BPSPL.03/PK.230/REKOM/X/2018.

## Tissue sample collection

From October 2017 to November 2018, a total of 100 tissue samples were obtained from *S. lewini*, including 41 from the fishing ports of Meulaboh and Aceh Jaya, 30 from a local shark landing in Manggar, and 29 from the fishing port of Tanjung Luar (Table 1). The samples (~0.5 cm$^3$) were dissected and preserved in sample bottles containing 96% ethanol.

## DNA extraction, amplification, and sequencing

DNA extraction was performed at the Biodiversity and Biosystematics Laboratory, IPB University, according to the protocol of the gSYNC DNA extraction kit product. A fragment of the mitochondrial cytochrome oxidase subunit 1 (*CO1*) gene was amplified using the forward primer fish-BCL (5'-TCA ACY AAT CAY AAA GAT ATY GGC AC-3') and the reverse fish-BCH (5'-ACT TCY GGG TGR CCR AAR AAT CA-3') [24, 25] in a 24 μL reaction mixture consisting of 3 μL of DNA template, 12.5 μL of MyTaq HS Red Mix, 9 μL of ddH$_2$O, 1.25 μL each of forward and reverse primers. Meanwhile, the reaction mixture was processed in a polymerase chain reaction (PCR) on a thermocycler using modified cycling conditions [26, 27]: pre-denaturation at 94˚C for 15 s. This process was followed by 38 cycles of denaturation at 94˚C for 30 s, annealing at 50˚C for 30 s, and extension at 72˚C for 45 s; as well as a final extension at 72˚C for 10 min. In addition, the amplicons were visualized by 1.5% agarose gel electrophoresis added with ethidium bromide at 100 V for 20 min. The gel was observed under UV light to identify bands showing the presence of DNA fragments. Sequencing was also performed using a machine with an optimized protocol of Sanger method [28].

All laboratory protocols on sampling and DNA identification methods were deposited in protocols.io platform with a digital object identifier (DOI) available at dx.doi.org/10.17504/protocols.io.bfwmjpc6.

## Data analysis

**Genetic diversity.** Over 179 mitochondrial *CO1* DNA sequences with an average length of 594 bp were edited and aligned using the ClustalW algorithm [29] implemented in MEGA 6.06 [30]. Genetic diversity parameters, such as the number of haplotypes and diversity (h) as well as nucleotide diversity (π), were calculated using the DNASp v6 [31] and Arlequin v.3.5 program [32]. Furthermore, additional *CO1* sequencing data of *S. lewini* from West Papua retrieved from GenBank (Table 2) (n = 14) were reanalyzed. These days were obtained by Sembiring *et al.* [33], sequences from previous studies performed in India (n = 6) [34], the United

**Table 1. Sampling collection sites at major shark landing sites in Indonesia.**

| Site | Geographic coordinate | Number of samples |
|---|---|---|
| **Aceh (ACH)** | | |
| Meulaboh | N 04˚ 08' 29" E 96˚ 07' 55" | 33 |
| Aceh Jaya | N 04˚ 38' 34" E 95˚ 34' 58" | 8 |
| **Balikpapan (BPN)** | | |
| Manggar | S 01˚ 12' 53" E 116˚ 58' 24" | 30 |
| **Lombok (LOM)** | | |
| Tanjung Luar | S 08˚ 46' 39" E 116˚ 31' 01" | 29 |
| **Total** | | 100 |

**Table 2. Localities, the total number (n), and accession number of *CO1* gene sequences of *S. lewini* from Aceh, Balikpapan, Lombok, and Western Papua (Indonesia), India, the United Arab Emirates, as well as Madagascar (Western Indian Ocean).**

| Locality | n | Accession number | Author |
|---|---|---|---|
| **Indonesia** | | | |
| Aceh | 41 | MT324149-156, MT324187-219 | This study |
| Balikpapan | 30 | MT324157-186 | This study |
| Lombok | 29 | MT324220-248 | This study |
| West Papua | 14 | KF590254-55, KF590271-76, KF793729, KF793738-42 | [33] |
| **Western Indian Ocean** | | | |
| India | 6 | KF899746-51 | [34] |
| United Arab Emirates | 30 | KP177238-41, KP177241, KP177254, KP177262, KP177272, KP177285-99, KP177300-07 | [35] |
| Madagascar | 29 | HQ171735-47, HQ171761-76 | [36] |

Arab Emirates (n = 30) [35], and Madagascar (n = 29) [36] to assess the genetic diversity in Indonesian and Western Indian Ocean populations.

**Population structure.** An analysis of molecular variance (AMOVA) and fixation index ($F_{ST}$) [37] was performed for three major groups: 1) within and among the four populations from Indonesia, 2) within and among the three populations from the Western Indian Ocean, and 3) comparison between populations from Indonesia and Western Indian Ocean using the Arlequin v.3.5 program (set up, 1000 permutations; significance level threshold, α = 0.05). These two analyses allowed the estimation of the overall extent of the genetic variation and differentiation level in Indonesia and the Western Indian Ocean. Furthermore, population differentiation and its significance between sampling sites were also calculated with pairwise estimates [38–40].

**Genetic connectivity.** A haplotype network was constructed with a median-joining method in the Network v5.1.1.0 program [41] for all haplotypes detected in Indonesia and the Western Indian Ocean. This network aimed to obtain haplotype connectivity following a broader spatial connection in the regional area of the Indian Ocean. The distribution of haplotypes for each location was also provided in a proper map to show the clear distributions and genetic connectivity among the populations.

## Results

### Genetic diversity

All sequences of *S. lewini* obtained were deposited in the BOLD System with the Barcode Index Number (BIN) registry of BOLD: AAA2403 and database of GenBank with accession numbers MT324149-248 (Table 2). A total of 179 sequences of 594 bp mitochondrial *CO1* gene was obtained from three sampling sites (Aceh, Balikpapan, and Lombok). Meanwhile, additional sequences of samples from West Papua and the Western Indian Ocean region were used to generate a total of 11 haplotype variations with 40 polymorphic loci (Table 3).

The comparison of the genetic diversity of *S. lewini* following haplotype and nucleotide diversity showed the presence of variation (Table 4). The haplotype diversity (h) among the samples obtained from Indonesia ranged from 0.143 to 0.668, while the nucleotide diversity (π) ranged from 0.000 to 0.020. The highest genetic diversity was observed for the samples

**Table 3. Forty polymorphic loci of 11 haplotypes from 179 sequences of the mitochondrial CO1 gene of *S. lewini* samples from four localities in Indonesia and three localities in the Western Indian Ocean region.**

| Haplotypes | Locus Position | | | | | | | | | | | | | | | | | | | | | | | | | | | | | | | | | | | | | | | |
|---|---|---|---|---|---|---|---|---|---|---|---|---|---|---|---|---|---|---|---|---|---|---|---|---|---|---|---|---|---|---|---|---|---|---|---|---|---|---|---|---|
| | 1 | 15 | 24 | 27 | 33 | 45 | 48 | 63 | 69 | 105 | 120 | 138 | 141 | 156 | 165 | 174 | 186 | 198 | 229 | 279 | 289 | 303 | 306 | 325 | 339 | 342 | 346 | 366 | 396 | 420 | 453 | 456 | 459 | 483 | 495 | 498 | 522 | 531 | 534 | 543 |
| H1* | A | T | A | C | T | T | G | T | T | A | T | C | G | T | C | C | C | T | T | C | G | T | T | C | C | A | C | T | T | C | C | T | T | C | T | C | C | T | C | T |
| H2* | . | . | . | A | . | . | . | . | . | G | . | A | T | . | . | T | T | C | C | T | . | C | C | T | T | C | . | C | C | T | . | . | . | A | . | T | . | C | . | C |
| H3 | . | . | . | A | . | . | . | . | C | G | . | A | T | . | . | T | T | C | C | T | . | C | C | T | T | C | . | C | C | T | . | . | . | A | . | T | . | C | . | . |
| H4* | . | . | . | A | . | . | . | . | . | G | . | A | T | . | . | T | T | C | C | T | . | C | C | T | T | C | . | C | C | T | . | . | . | A | . | T | . | C | . | . |
| H5 | . | . | . | . | . | . | . | . | . | . | . | . | . | . | . | . | . | . | . | . | . | . | . | . | . | . | . | . | . | . | . | . | G | . | G | . | . | . | . | . |
| H6 | . | . | . | . | . | C | . | . | . | . | . | . | . | . | . | . | . | . | . | . | . | . | . | . | . | . | . | . | . | . | . | . | . | . | . | . | . | . | . | . |
| H7 | . | . | C | . | . | . | . | . | . | . | . | . | . | G | . | . | . | . | . | . | . | . | . | . | . | . | . | . | . | . | . | . | . | . | . | . | . | . | . | . |
| H8 | . | C | . | . | C | . | . | . | . | . | . | . | . | . | . | . | . | . | . | . | . | . | . | . | . | . | . | . | . | . | . | . | . | . | . | . | . | . | . | . |
| H9 | . | . | . | . | . | . | . | . | . | . | G | . | . | . | . | . | . | G | . | . | T | . | . | . | . | . | . | . | . | . | . | . | . | . | . | . | . | . | G | . |
| H10 | G | . | . | . | . | . | . | G | . | . | . | . | . | G | . | . | . | . | . | . | . | . | . | . | . | . | . | . | . | . | . | . | . | . | . | . | . | . | . | . |
| H11 | . | . | . | . | . | . | . | . | . | . | . | . | . | . | . | . | . | . | . | . | . | . | . | . | . | . | T | . | . | . | T | . | . | . | C | . | T | . | . | . |

Notes:

Nucleobase at each position is given for H1 while those different are written for all other haplotypes. Nucleobases identical to H1 are indicated with dots (.).

*Three original haplotypes of the *S. lewini* populations were obtained from the areas of study (Aceh, Balikpapan, and Lombok) in Indonesia. The remaining haplotypes were reanalyzed from previous studies.

**Table 4. Genetic diversity of *S. lewini* based on sample size (n), haplotype number (Hn), haplotype diversity (h), and nucleotide diversity (π) in samples from each site in Indonesia and the Western Indian Ocean region.**

| Population | n | Genetic Diversity | | |
|---|---|---|---|---|
| | | Hn | h | Π |
| **Indonesia** | | | | |
| Aceh (ACH) | 41 | 4 | 0.668 | 0.020 |
| Balikpapan (BPN) | 30 | 3 | 0.646 | 0.002 |
| Lombok (LOM) | 29 | 3 | 0.362 | 0.001 |
| West Papua (WEP) | 14 | 2 | 0.143 | 0.000 |
| Overall Indonesia | 114 | 4 | 0.717 | 0.013 |
| **Western Indian Ocean** | | | | |
| India (IND) | 6 | 1 | 0.000 | 0.000 |
| United Arab Emirates (UAE) | 30 | 8 | 0.467 | 0.002 |
| Madagascar (MDG) | 29 | 1 | 0.000 | 0.000 |
| Overall Western Indian Ocean | 65 | 8 | 0.232 | 0.001 |

from Aceh (h = 0.668; π = 0.020), followed by the Balikpapan population, which exhibited a lower haplotype and nucleotide diversity (h = 0.646; π = 0.002). On the contrary, the lowest genetic diversity was detected in West Papua (h = 0.143, π = 0.000). Similarly, the *S. lewini* population from Lombok exhibited a fairly low genetic diversity (h = 0.362; π = 0.001). However, the overall diversity in Indonesia was relatively high (h = 0.717; π = 0.013) since the average in the Western Indian Ocean region was low and ranged from 0.000 to 0.467. Therefore, it is reasonable to conclude that the overall diversity in the Western Indian Ocean region was also low (h = 0.232; π = 0.001).

## Population structure

The analysis of the fixation index ($F_{ST}$) and the corresponding *P*-values between and within the four *S. lewini* populations (ACH, BPN, LOM, and WEP) from Indonesia and three populations (IND, UAE, and MDG) from the Western Indian Ocean region are shown in Table 5. The overall $F_{ST}$ value in Indonesia was significantly higher than that observed in other regions ($F_{ST}$ = 0.442; $P < 0.001$) due to the presence of multiple subdivisions.

**Table 5. Analysis of molecular variance (AMOVA) for the percentage of variation (%), $F_{ST}$ value, and significance level (*P*-value) in *S. lewini* samples from Indonesian, the Western Indian Ocean, and between Indonesian and Western Indian Ocean populations.**

| Source of variation | df | Percentage of variation (%) | $F_{ST}$ value | *P*-value |
|---|---|---|---|---|
| **Indonesia** | | | | |
| Among Populations | 3 | 44.15 | 0.442 | 0.000 |
| Within Populations | 110 | 55.85 | | |
| Total | 113 | | | |
| **Western Indian Ocean** | | | | |
| Among Populations | 2 | −1.31 | −0.013 | 0.612 |
| Within Populations | 62 | 101.31 | | |
| Total | 64 | | | |
| **Indonesia vs. Western Indian Ocean** | | | | |
| Among Populations | 1 | 74.04 | 0.740 | 0.000 |
| Within Populations | 177 | 25.96 | | |
| **Total** | 178 | | | |

**Table 6. Pairwise $F_{ST}$ values (below the diagonal) and $P$-values (above the diagonal) between the *S. lewini* populations from Aceh (ACH), Balikpapan (BPN), Lombok (LOM), West Papua (WEP), and Western Indian Ocean (WIO).**

| Sample Sites | ACH | BPN | LOM | WEP | WIO |
|---|---|---|---|---|---|
| ACH | - | 0.000 | 0.000 | 0.000 | 0.000 |
| BPN | 0.427 | - | 0.091 | 0.000 | 0.000 |
| LOM | 0.438 | 0.044 | - | 0.000 | 0.000 |
| WEP | 0.398 | 0.495 | 0.736 | - | 0.000 |
| WIO | 0.509 | 0.965 | 0.973 | 0.975 | - |

A genetic homogeneity was observed in the population from the Western Indian Ocean region ($F_{ST} = -0.013$; $P = 0.612$). However, a comparison of the population structure between Indonesia and the Western Indian Ocean region yielded significant differentiation ($F_{ST} = 0.740$; $P < 0.001$). The pairwise $F_{ST}$ values between the populations from the four locations in Indonesia and the Western Indian Ocean population are shown in Table 6. Furthermore, the overall pairwise analysis through the distance method showed the presence of significant differentiation among the four populations. In contrast, BPN and LOM ($F_{ST} = 0.044$; $P = 0.091$) showed fairly low $F_{ST}$ values and no significant $P$-values. Furthermore, among the populations from Indonesia, the ACH showed a trend of being closer to the Western Indian Ocean since it exhibited a lower $F_{ST}$ and significant $P$-value ($F_{ST} = 0.509$; $P < 0.001$).

## Genetic connectivity

Network analysis of the haplotype identified two main groups of haplotypes (Fig 1) referred to as clade A and B. Clade A consisted of haplotype H1, which was observed in several regions, i.e., Aceh, India, United Arab Emirates, and Madagascar, while H5, H6, H7, H8, H9, H10, and H11 were present only in samples from the United Arab Emirates. Clade B consisted of three haplotypes (H2, H3, and H4), which were spread evenly in Indonesia. Haplotypes H1, H3, and H4 were predominant in Aceh, West Papua, and Balikpapan-Lombok, respectively (Fig 2).

## Discussion

### Genetic diversity

The overall genetic diversity of *S. lewini* at the haplotype and nucleotide levels was relatively high for populations in Indonesia. These findings are consistent with the results reported by Ovenden *et al.* [23] regarding the mitochondria control region from three localities in Indonesia [22]. The scalloped hammerhead sharks are a highly migratory species with a wide distribution in tropical and warm-temperate waters. This specie can move across oceanic waters to a distance of up to 1671 km [12]. Due to its migratory ability and broad ecological niches, this species tends to have higher genetic diversity than others [42]. Generally, high levels of genetic diversity are associated with large population size [43] and are promoted by several factors, such as local population sizes, fast generation times [44], high nucleotide substitution rates [45], and high gene flow between geographically distant populations.

The finding of a relatively high genetic diversity for *S. lewini* appears to be inconsistent on the assumption that overexploitation of this species as both a target of fishing and bycatch led to the decline of its populations on a global scale [46]. However, the results obtained from Lombok may be relevant since the lower genetic diversity detected was probably driven by continuous fishing pressure. *S. lewini* species are the top three targeted sharks at the Tanjung

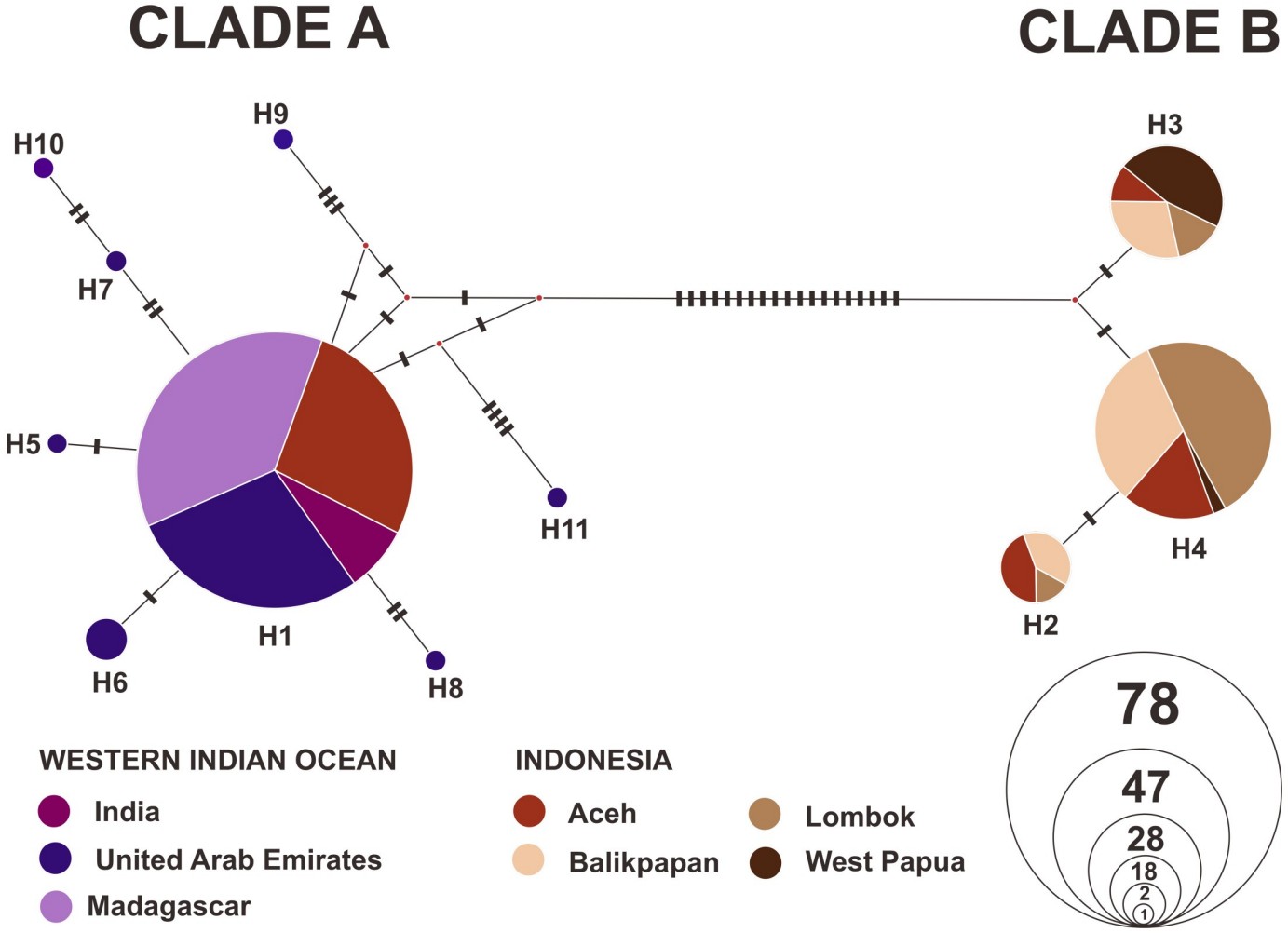

**Fig 1. Haplotype network of the *S. lewini* (n = 179) population from Indonesia and the Western Indian Ocean region, which was constructed using the median joining method.**

Luar fishing port in Lombok and have faced high fishing pressure over more than 40 years with recent exploitation rates (E) reaching 0.59 [47].

Furthermore, according to the global fisheries information system on a global scale and Indonesia by FAO [48], the hammerhead sharks (Sphyrnidae), including *S. lewini*, are very important. These species were highly exploited in the last two decades, with an estimated rapid increase in global capture, from 220 tons per year in 1985 up to approximately 10,362 in 2016. During the same period, the capture level also increased significantly, reaching approximately 1,492 tons in 2016. Meanwhile, Indonesia recorded one of the highest numbers of sharks and rays caught on the global catches reported in 2000–2011 [49].

Clarke *et al*. [50] reported similar findings regarding the mitochondrial DNA of the silky shark, *Carcharhinus falciformis*, which exhibited a high genetic diversity under circumstances of overexploitation. Elasmobranchs exhibit adaptability to environmental and anthropogenic stresses, which causes genetic bottlenecks because of their particular life histories [8]. However, the population decline caused by recent fishery activities might be insufficient to reduce genetic diversity, particularly for species with a long life span (13–20 years), such as *S. lewini* [51, 52].

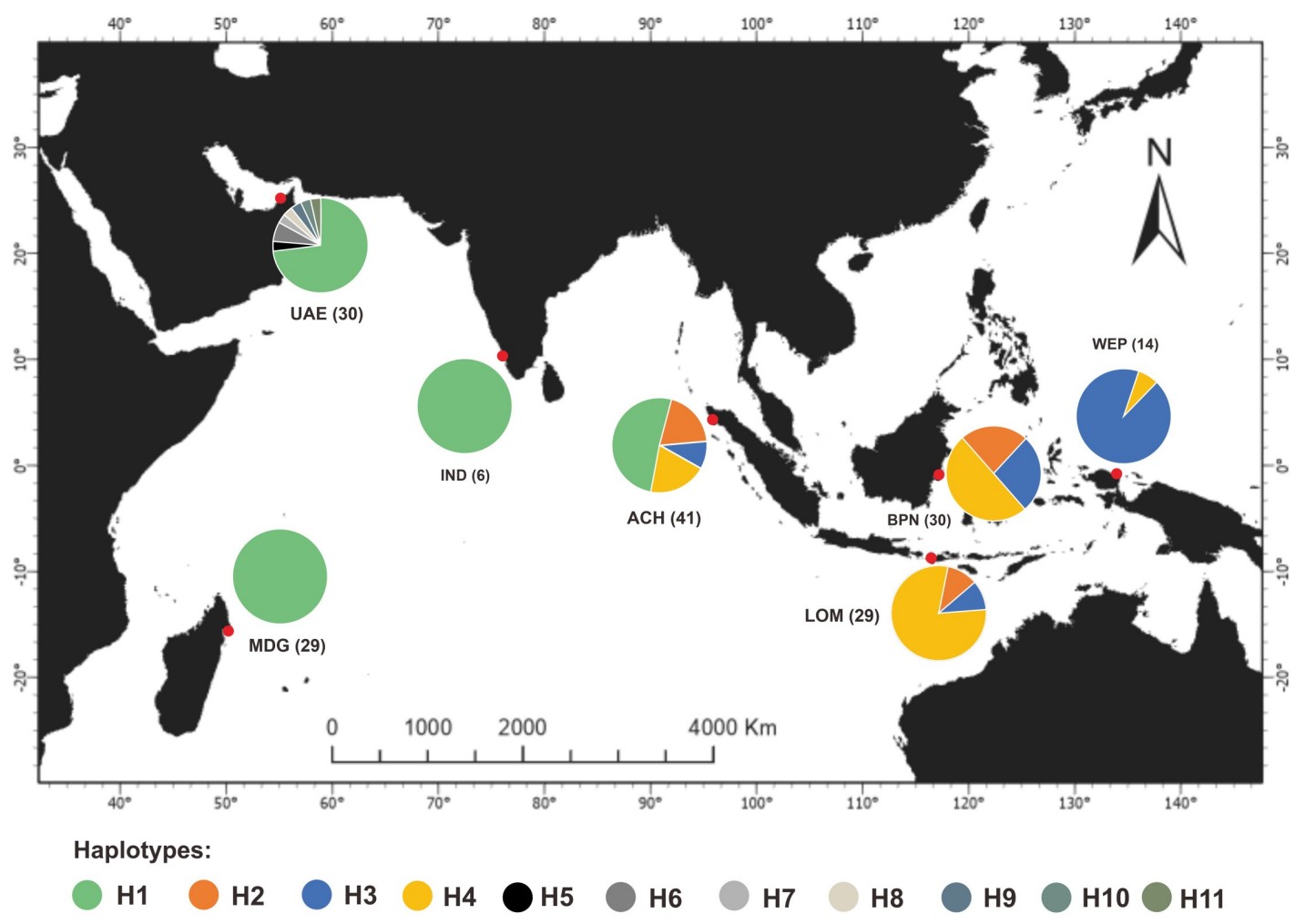

**Fig 2. Distribution of the 11 haplotypes of the *S. lewini* population from Indonesia and Western Indian Ocean at the regional scale.**

The persistent decline, as predicted for the Lombok population of *S. lewini*, correlated positively with the loss of genetic diversity and created a bottleneck [52, 53], as reported by Pinsky and Palumbi in a meta-analysis of several marine fish [54].

## Population structure

The obtained results showed the presence of homogeneity among the *S. lewini* populations from Balikpapan and Lombok. The pairwise $F_{ST}$ analysis detected no significant genetic differentiation in these two populations with the lowest value. These findings complement previous studies conducted in Indo-Australian waters. Similarly, Ovenden *et al.* [23] reported evidence for the mitochondrial control region regarding the structure, with no differentiation between two populations of *S. lewini* (Lombok and northern Australia). This pattern of single-stock population suggests that these localities are a migration zone of *S. lewini* and a reproductive movement may occur in their coastal areas. However, there are strong Indonesian through flow currents between Kalimantan and Sulawesi Island. Adult *S. lewini* specimens are highly migratory, with a large body supporting the high dispersal ability of this species. Consequently, they show the possibility of overcoming that geographical barrier. In contrast, different results

from the comparison among these two populations from central Indonesia and one from Aceh (western Indonesia) as well as from West Papua (eastern Indonesia), which exhibited strong genetic differentiation were obtained. Furthermore, these regions were spatially separated by a long distance, with the possibility of the existence of more complex barriers. These barriers can be inter-island or anthropogenic factors on a high commercial and artisanal fishing pressure along the southern and northern coasts of Java.

## Genetic connectivity

Regarding the significant value of $F_{ST}$ among the population ($F_{ST}$ = 0.740; $P$ < 0.001) as well as haplotype network and distribution shown in Figs 1 and 2, two restricted haplogroups which separated the *S. lewini* population in Indonesian with the Western Indian Ocean population were observed. The two haplogroups were separated with 19 different nucleotide bases due to monomorphic and polymorphic mutations. Furthermore, limited gene flow that occurs among populations in Indonesia forms a different pool with the Western Indian Ocean population. However, this was expected because of the complex geographic barrier in Indonesia's marine ecosystem, and the global distribution pattern of *S. lewini*, with significant separation population across ocean basin as well as discontinuous coastline habitat [6].

Generally, *S. lewini* populations from Balikpapan, Lombok, and West Papua appear to be isolated from the Western Indian Ocean and shared a haplotype network exclusively only in eastern Indonesian waters. However, an interesting result regarding genetic sharing between the populations from Aceh and the Indian Ocean population was obtained. H1 is a unique haplotype that was only obtained in Aceh. However, the $F_{ST}$ value observed significant differences between the population in Aceh and the Western Indian Ocean, and there was an indication of genetic sharing between those localities ($F_{ST}$ = 0.509; $P$ < 0.001). The similarity between the predominant haplotype (H1) of *S. lewini* from Aceh and that of the populations from India, the United Arab Emirates as well as Madagascar reflected a genetic sharing process in the Indian Ocean region. This showed the presence of past historical gene flow between the populations in spatially separated regions driven by ancestral interaction [17]. However, recent studies reported that the scalloped hammerhead demonstrated a strong differentiation in population structure across ocean basins e.g. Indian Ocean and discontinuous continental coastlines, as shown by the separation between Aceh and Indian coastline [6, 22].

## Conservation implications

The high diversity of the *S. lewini* populations in Indonesia shows that this species has not experienced a genetic loss because of exploitation pressure. However, the lower genetic diversity of *S. lewini* from Lombok and West Papua showed a higher risk of loss, which probably was the result of high fisheries pressure. Furthermore, the genetic assessment of *S. lewini* samples from four localities showed that a single stock exists between Lombok and Balikpapan. On the contrary, a separate stock was observed for Aceh and West Papua, showing that the management of this species should occur on a stock-based approach at least on three mitochondrial-stock conservation units. The complex population of *S. lewini* with limited connectivity observed in Indonesia and the Western Indian Ocean region demonstrated the importance of promoting specific collaborative management strategies among Indonesian, and in conjunction with Western Indian Ocean agencies at the regional scale.

## Conclusion

This study provided important findings on the population structure of *S. lewini* in Indonesia, with a high genetic diversity and three significant subdivisions. The results showed the

capability of the population to adapt to rapid environmental changes and pressure, including fishing activities. In addition, the lower genetic diversity in Lombok and West Papua was also considered. The restricted genetic sharing detected among the species obtained from Indonesia showed unique features among these populations. Therefore, a specific collaborative action across regions is needed to promote sustainable management and conservation purposes, both in Indonesia and at the regional scale in the Western Indian Ocean area.

## Acknowledgments

The authors are grateful to the institutions and individuals that have made the study possible: colleagues from Marine Biodiversity and Biosystematics Lab (BIODIVSI), Faculty of Marine Science and Technology, IPB University for their help during this study. The authors are further grateful to BPSPL Satker Balikpapan, Listian Nova, and Mr. Hery, Head of TPI Manggar, Balikpapan for their help during the sample's collection, and to reviewers and proofreaders that have provided highly constructive suggestions for better writing of the study.

## Author Contributions

**Conceptualization:** Sutanto Hadi, Efin Muttaqin, Benaya M. Simeon, Muhammad Ichsan, Beginer Subhan, Hawis Madduppa.

**Data curation:** Efin Muttaqin, Benaya M. Simeon, Muhammad Ichsan.

**Formal analysis:** Sutanto Hadi.

**Funding acquisition:** Noviar Andayani.

**Investigation:** Sutanto Hadi.

**Methodology:** Sutanto Hadi, Efin Muttaqin, Benaya M. Simeon, Muhammad Ichsan, Hawis Madduppa.

**Project administration:** Hawis Madduppa.

**Resources:** Efin Muttaqin, Benaya M. Simeon, Muhammad Ichsan, Hawis Madduppa.

**Software:** Sutanto Hadi.

**Supervision:** Noviar Andayani, Beginer Subhan, Hawis Madduppa.

**Validation:** Sutanto Hadi.

**Visualization:** Sutanto Hadi.

**Writing – original draft:** Sutanto Hadi, Hawis Madduppa.

**Writing – review & editing:** Noviar Andayani, Hawis Madduppa.

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
