## [Editor Report · Decision Letter 0]

18 Mar 2020

PONE-D-20-06557

Genetic connectivity and diversity of endangered species the scalloped hammerhead shark Sphyrna lewini (Griffith & Smith 1834) population in Indonesia and Western Indian Ocean

PLOS ONE

Dear Dr. Madduppa,

Thank you for submitting your manuscript to PLOS ONE. After careful consideration, we feel that it has merit but does not fully meet PLOS ONE’s publication criteria as it currently stands. Therefore, we invite you to submit a revised version of the manuscript that addresses the points raised during the review process.

This is a very interesting manuscript and it will be definitely useful in the future for comparisons for example in market surveys.  However, overall the quality of the english needs to be improved.  I think once this has been taken care of other details regarding presentation of results, etc will be much clearer to review.  Also, figures need to be improved, since resolution is low and they look grainy.  Attached you will find a file with some suggestions.  I work on this particular subject (conservation genetics of aquatic vertebrates) and I have a lot of experience in sharks, so I can really see the potential in this manuscript, but english needs to be improved a lot.

We would appreciate receiving your revised manuscript by 20th of April. To enhance the reproducibility of your results, we recommend that if applicable you deposit your laboratory protocols in protocols.io, where a protocol can be assigned its own identifier (DOI) such that it can be cited independently in the future. For instructions see: http://journals.plos.org/plosone/s/submission-guidelines#loc-laboratory-protocols

We look forward to receiving your revised manuscript.

Kind regards,

Susana Caballero, PhD

Academic Editor

PLOS ONE

Additional Editor Comments (if provided):

This is a very interesting manuscript and it present very important information that will help understanding the genetic diversity of scalloped hammerhead in this region and this information will be likely used for future assignment comparisons for example for market surveys etc. However, the english NEEDS to be improved. I really hope the authors can find the help of a native english speaker that can help them with this. I wrote some things in the PDF I am attaching but it definitely needs an overall improvement of the english.

The resolution of the figures needs to be improved because they look grainy.

Journal Requirements:

2. In your Methods section, please provide additional location information of the sampling collection sites, including geographic coordinates for the data set if available.

"This research was supported by Wildlife Conservation Society (WCS)-Indonesia, in collaboration with Faculty of Fisheries and Marine Sciences IPB University. Sutanto Hadi received a scholarship from Indonesia Endowment Fund for Education (LPDP)."

"NO - The funders had no role in study design, data collection and analysis, decision to publish, or preparation of the manuscript."
---

## [Author Response · Author response to Decision Letter 0]

7 Apr 2020

Dear Susana Caballero, PhD

Academic Editor

PLOS ONE

Thank you for reviewing and suggesting our manuscript:

PONE-D-20-06557

Genetic connectivity and diversity of endangered species the scalloped hammerhead shark Sphyrna lewini (Griffith & Smith 1834) population in Indonesia and Western Indian Ocean

PLOS ONE

I would like to response your comments:

Additional Editor Comments (if provided):

1. This is a very interesting manuscript and it present very important information that will help understanding the genetic diversity of scalloped hammerhead in this region and this information will be likely used for future assignment comparisons for example for market surveys etc. However, the english NEEDS to be improved. I really hope the authors can find the help of a native english speaker that can help them with this. I wrote some things in the PDF I am attaching but it definitely needs an overall improvement of the english.

Thank you for your interest and suggestion. We have improved overall of the English for better manuscript after proof read by native English speaker.

2. The resolution of the figures needs to be improved because they look grainy.

Thank you for your suggestion. We have improved the resolution of figures and generated in PACE to meet PLOS ONE requirement.

We would appreciate receiving your revised manuscript by 20th of April. To enhance the reproducibility of your results, we recommend that if applicable you deposit your laboratory protocols in protocols.io, where a protocol can be assigned its own identifier (DOI) such that it can be cited independently in the future. For instructions see: http://journals.plos.org/plosone/s/submission-guidelines#loc-laboratory-protocols

• A rebuttal letter that responds to each point raised by the academic editor and reviewer(s). This letter should be uploaded as separate file and labeled 'Response to Reviewers'.

• A marked-up copy of your manuscript that highlights changes made to the original version. This file should be uploaded as separate file and labeled 'Revised Manuscript with Track Changes'.

• An unmarked version of your revised paper without tracked changes. This file should be uploaded as separate file and labeled 'Manuscript'.

Journal Requirements:

We have read and followed the format as The PLOS ONE style templates

2. In your Methods section, please provide additional location information of the sampling collection sites, including geographic coordinates for the data set if available.

We have provided additional location information including geographic coordinates in a table in manuscript

We have asked our colleague to check and edit our manuscript,

 "This research was supported by Wildlife Conservation Society (WCS)-Indonesia, in collaboration with Faculty of Fisheries and Marine Sciences IPB University. Sutanto Hadi received a scholarship from Indonesia Endowment Fund for Education (LPDP)."

"NO - The funders had no role in study design, data collection and analysis, decision to publish, or preparation of the manuscript."

Thank you for your suggestion. We have removed any funding-related statement from manuscript.

5. PLOS requires an ORCID iD for the corresponding author in Editorial Manager on papers submitted after December 6th, 2016. Please ensure that you have an ORCID iD and that it is validated in Editorial Manager. To do this, go to ‘Update my Information’ (in the upper left-hand corner of the main menu), and click on the Fetch/Validate link next to the ORCID field. This will take you to the ORCID site and allow you to create a new iD or authenticate a pre-existing iD in Editorial Manager. Please see the following video for instructions on linking an ORCID iD to your Editorial Manager account:

https://www.youtube.com/watch?v=_xcclfuvtxQ

Reviewers' comments:

Thank you for your suggestion. We have uploaded and adjusted the pictures to PACE to meet PLOS ONE figures requirement. We attached the picture in submission.

---

## [Editor Report · Decision Letter 1]

10 Apr 2020

PONE-D-20-06557R1

Genetic connectivity of the scalloped hammerhead shark Sphyrna lewini (Griffith & Smith 1834) population across Indonesia and Western Indian Ocean

PLOS ONE

Dear Dr. Maddupa,

Thank you for submitting your manuscript to PLOS ONE. After careful consideration, we feel that it has merit but does not fully meet PLOS ONE’s publication criteria as it currently stands. Therefore, we invite you to submit a revised version of the manuscript that addresses the points raised during the review process.

There are still a number of grammatical english mistakes along the manuscript that need to be corrected.  I would suggest looking into a service that revises manuscripts. sometimes local Universities have this service in place in language schools.  I can see some mistakes for example in the results section, when you present results from the AMOVA.  This is very important because these results need to be very clear for the readers.

Also, I think it is good to submit your sequences not only to BOLD but also to Genbank, since for genetic studies, it is high likely that researchers will look for information for comparisons on Genbank.

We would appreciate receiving your revised manuscript by 10 of May, particularly considering the current situation regarding Covid-19. To enhance the reproducibility of your results, we recommend that if applicable you deposit your laboratory protocols in protocols.io, where a protocol can be assigned its own identifier (DOI) such that it can be cited independently in the future. For instructions see: http://journals.plos.org/plosone/s/submission-guidelines#loc-laboratory-protocols

We look forward to receiving your revised manuscript.

Kind regards,

Susana Caballero, PhD

Academic Editor

PLOS ONE

Additional Editor Comments (if provided):

There are still a number of grammatical errors. I think it is important to have a service look over this manuscript.

Please submit the sequences you generated to Genbank as well as BOLD, since probably more people will have an opportunity to look for these sequences in this database

---

## [Author Response · Author response to Decision Letter 1]

25 Apr 2020

Dear Susana Caballero, PhD

Academic Editor

PLOS ONE

Thank you for reviewing and suggesting our manuscript:

“Genetic connectivity and diversity of endangered species the scalloped hammerhead shark Sphyrna lewini (Griffith & Smith 1834) population across Indonesia and the Western Indian Ocean”

I would like to response your comments:

Additional Editor Comments (if provided):

This is a very interesting manuscript and it present very important information that will help understanding the genetic diversity of scalloped hammerhead in this region and this information will be likely used for future assignment comparisons for example for market surveys etc. However, the English NEEDS to be improved. I really hope the authors can find the help of a native English speaker that can help them with this. I wrote some things in the PDF I am attaching but it definitely needs an overall improvement of the English

.

Thank you for your interest and suggestion. We have improved overall of the English for better manuscript after proof read by professional native English speaker.

The resolution of the figures needs to be improved because they look grainy.

Thank you for your suggestion. We have improved the resolution of figures and generated in PACE to meet PLOS ONE requirement.

We would appreciate receiving your revised manuscript by 20th of April. To enhance the reproducibility of your results, we recommend that if applicable you deposit your laboratory protocols in protocols.io, where a protocol can be assigned its own identifier (DOI) such that it can be cited independently in the future. For instructions see: ttp://journals.plos.org/plosone/s/submission-guidelines#loc laboratory-protocols

• A rebuttal letter that responds to each point raised by the academic editor and reviewer(s). This letter should be uploaded as separate file and labelled 'Response to Reviewers'.

• A marked-up copy of your manuscript that highlights changes made to the original version. This file should be uploaded as separate file and labelled 'Revised Manuscript with Track Changes'.

•An unmarked version of your revised paper without tracked changes. This file should be uploaded as separate file and labelled 'Manuscript'.

Journal Requirements:

1. Please ensure that your manuscript meets PLOS ONE's style requirements, including those for file naming. The PLOS ONE style templates can be found at http://www.plosone.org/attachments/PLOSOne_formatting_sample_main_body.pdf and

http://www.plosone.org/attachments/PLOSOne_formatting_sample_title_authors_affiliat

ions.pdf

Thank you for your suggestion. We have read and followed the format as The PLOS ONE style templates.

2. In your Methods section, please provide additional location information of the sampling collection sites, including geographic coordinates for the data set if available.

Thank you for your suggestion. We have provided additional location information including geographic coordinates in a table in manuscript

Thank you for your suggestion. We have sent and proof read our manuscript to a professional scientific editor to check and enhance our manuscript grammar, spelling and language use.

"This research was supported by Wildlife Conservation Society (WCS)-Indonesia, in collaboration with Faculty of Fisheries and Marine Sciences IPB University. Sutanto Hadi received a scholarship from Indonesia Endowment Fund for Education (LPDP)."

"NO - The funders had no role in study design, data collection and analysis, decision to

publish, or preparation of the manuscript."

Thank you for your suggestion. We have removed any funding-related statement from the

manuscript.

Please see the following video for instructions on linking an ORCID iD to your Editorial Manager account:

https://www.youtube.com/watch?v=_xcclfuvtxQ

Reviewers' comments:

Thank you for your suggestion. We have uploaded and adjusted the pictures to PACE to meet PLOS ONE figures requirement. We attached the picture in submission.

Please send them (sequence data) to GenBank as well.

Thank you for your suggestion. We have deposited our all sequences data to Genbank as well with accession number MT324149 to MT324248.

---

## [Editor Report · Decision Letter 2]

30 Apr 2020

PONE-D-20-06557R2

Genetic connectivity of the scalloped hammerhead shark Sphyrna lewini (Griffith & Smith 1834) population across Indonesia and the Western Indian Ocean

PLOS ONE

Dear  Dr.Madduppa,

Thank you for submitting your manuscript to PLOS ONE. After careful consideration, we feel that it has merit but does not fully meet PLOS ONE’s publication criteria as it currently stands. Therefore, we invite you to submit a revised version of the manuscript that addresses the points raised during the review process.

The English still needs further improvement and I think the authors need to further analyze their genetic connectivity section.  In the discussion, it does not reflect some of the findings you get, particularly from your haplotype network.  Please read my suggestions on the attached PDF.

We would appreciate receiving your revised manuscript by 15th May 2020. To enhance the reproducibility of your results, we recommend that if applicable you deposit your laboratory protocols in protocols.io, where a protocol can be assigned its own identifier (DOI) such that it can be cited independently in the future. For instructions see: http://journals.plos.org/plosone/s/submission-guidelines#loc-laboratory-protocols

We look forward to receiving your revised manuscript.

Kind regards,

Susana Caballero, PhD

Academic Editor

PLOS ONE

Additional Editor Comments (if provided):

This is a very interesting paper, but I am still finding mistakes in the english. Also, the authors need to do further interpretation of their results, particularly regarding the population connectivity. I am including the text with my suggestions.

---

## [Author Response · Author response to Decision Letter 2]

7 May 2020

To:

Susana Caballero, PhD

Academic Editor

PLOS ONE

Thank you for reviewing and suggesting our manuscript:

“Genetic connectivity and diversity of endangered species the scalloped hammerhead shark Sphyrna lewini (Griffith & Smith 1834) population across Indonesia and the Western Indian Ocean”

I would like to response your comments in the following systematic table format:

1st SUBMISSION RESPONSE

No. REVIEWER COMMENT AUTHOR RESPONSES

 Additional Editor Comments

(if provided) 

1. This is a very interesting manuscript and it present very important information that will help understanding the genetic diversity of scalloped hammerhead in this region and this information will be likely used for future assignment comparisons for example for market surveys etc. However, the English NEEDS to be improved. I really hope the authors can find the help of a native English speaker that can help them with this. I wrote some things in the PDF I am attaching but it definitely needs an overall improvement of the English

 Thank you for your interest and suggestion. We have improved overall of the English for better manuscript after proof read by professional native English speaker.

2. The resolution of the figures needs to be improved because they look grainy Thank you for your suggestion. We have improved the resolution of figures and generated in PACE to meet PLOS ONE requirement.

3. We would appreciate receiving your revised manuscript by 20th of April. 4. To enhance the reproducibility of your results, we recommend that if applicable you deposit your laboratory protocols in protocols.io, where a protocol can be assigned its own identifier (DOI) such that it can be cited independently in the future. For instructions see: ttp://journals.plos.org/plosone/s/submission-guidelines#loc laboratory-protocols

• A rebuttal letter that responds to each point raised by the academic editor and reviewer(s). This letter should be uploaded as separate file and labelled 'Response to Reviewers'.

• A marked-up copy of your manuscript that highlights changes made to the original version. This file should be uploaded as separate file and labelled 'Revised Manuscript with Track Changes'.

•An unmarked version of your revised paper without tracked changes. This file should be uploaded as separate file and labelled 'Manuscript'.

No. Journal Requirements Author Responses

Please ensure that your manuscript meets PLOS ONE's style requirements, including those for file naming. The PLOS ONE style templates can be found at http://www.plosone.org/attachments/PLOSOne_formatting_sample_main_body.pdf and http://www.plosone.org/attachments/PLOSOne_formatting_sample_title_authors_affiliat

ions.pdf

 Thank you for your suggestion. We have read and followed the format as The PLOS ONE style templates.

2. In your Methods section, please provide additional location information of the sampling collection sites, including geographic coordinates for the data set if available.

 Thank you for your suggestion. We have provided additional location information including geographic coordinates in a table in manuscript.

 Thank you for your suggestion. We have sent and proof read our manuscript to a professional scientific editor to check and enhance our manuscript grammar, spelling and language use.

"This research was supported by Wildlife Conservation Society (WCS)-Indonesia, in collaboration with Faculty of Fisheries and Marine Sciences IPB University. Sutanto Hadi received a scholarship from Indonesia Endowment Fund for Education (LPDP)."

"NO - The funders had no role in study design, data collection and analysis, decision to

publish, or preparation of the manuscript."

 Thank you for your suggestion. We have removed any funding-related statement from the

manuscript.

6. PLOS requires an ORCID iD for the corresponding author in Editorial Manager on papers submitted after December 6th, 2016. Please ensure that you have an ORCID iD and that it is validated in Editorial Manager. To do this, go to ‘Update my Information’ (in the upper left-hand corner of the main menu), and click on the Fetch/Validate link next to the ORCID field. This will take you to the ORCID site and allow you to create a new iD or authenticate a pre-existing iD in Editorial Manager.

Please see the following video for instructions on linking an ORCID iD to your Editorial Manager account:

https://www.youtube.com/watch?v=_xcclfuvtxQ

2nd SUBMISSION RESPONSE

No. Reviewers' comments Author Responses

 Thank you for your suggestion. We have uploaded and adjusted the pictures to PACE to meet PLOS ONE figures requirement. We attached the picture in submission.

2. which populations have been declining worldwide Thank you for your correction, we have revised the sentence

3. use h instead of Hd Thank you for your correction, we have replaced all of term “Hd” with “h”

4. is considered coastal due to their need for nursery areas Thank you for your suggestion, we used term “semi-oceanic” as stated in some journals which have the similar meaning that you suggested, so we agreed and have revised the sentence

5. shorter phrases (Line 47-51) Thank you for your correction. We have shortened the phrases.

6. were already dead when collected Thank you for your correction. We have revised the sentence.

7. sample (Line 83) Thank you for your correction

8. 100 samples of which length?

 Thank you for your question, we used 594 bp of nucleotide and 179 sequences in total, not 100. We have revised it in the sentence

9. you previously mention a 100 (sequences)

 Thank you for your correction. We used 179 sequences in total, we have revised in previous statement

10. please send them (sequence data) to genbank as well Thank you for your suggestion. We have deposited our all sequences data to Genbank as well with accession number MT324149 to MT324248.

11. use identifiers particular for your area of study so that they dont get confused with other previously published Thank you for your suggestion. We used * (asterisk) as identifiers of original haplotype from area of study at notes section below the table.

12. repeat “find and finding” (Line 290)

 Thank you for your correction. We have revised the sentence.

 remove any funding-related text from the manuscript Thank you for your correction. We have removed any funding-related text.

No. Reviewers' comments Author Responses

 2nd Submission 

1. like before: usually 13-16 words,

title is too long Thank you for your correction. We have tried to shorter the title

2. which will make co-management actions necessary across

regions (Line 38-39) Thank you for your correction in Line 38-39.

3. I think the introduction too short.

According to your title you could write some more about the

management used now and what are the points of

improvement or change in your eyes for S. lewini Thank you for your correction. We have tried to enrich the introduction with what this study can give suggestion for better shark management and conservation

 What?

..additional.. (Line 177) Thank you for your correction in Line 177.

 What..? were displayed? That’s not a

sentence here (Line 181) in Table Thank you for your suggestion. We used * (asterisk) as identifiers of original haplotype from area of study at notes section below table.

 One or more? Than add an s Thank you for your correction. We have revised the sentence.

 One or more? Add s Thank you for your correction. We have revised the sentence.

 What?? Not a sentence here.. (Line 280) Thank you for your correction. We have revised the sentence.

 Here smaller font size 11 before 12!

check Thank you for your correction. We have fixed the font size.

 Check fond size here again its

smaller!! (Line 290,291,302) Thank you for your correction. We have fixed the font size.

 One or more? Add s Thank you for your correction.

 Strange phrase here..I dont know

what that should mean..

 Thank you for your correction. We have revised the sentence.

3rd SUBMISSION RESPONSE

No. Additional Editor Comments

(if provided) Author Responses

1. Thank you for submitting your manuscript to PLOS ONE. After careful consideration, we feel that it has merit but does not fully meet PLOS ONE’s publication criteria as it currently stands. Therefore, we invite you to submit a revised version of the manuscript that addresses the points raised during the review process. 

2. The English still needs further improvement and I think the authors need to further analyze their genetic connectivity section. In the discussion, it does not reflect some of the findings you get, particularly from your haplotype network. Please read my suggestions on the attached PDF. Thank you for your correction. We have added some sentences in discussion to reflect our genetic data particularly in connectivity section.

3. We would appreciate receiving your revised manuscript by 15th May 2020. 4. 5. To enhance the reproducibility of your results, we recommend that if applicable you deposit your laboratory protocols in protocols.io, where a protocol can be assigned its own identifier (DOI) such that it can be cited independently in the future. For instructions see: http://journals.plos.org/plosone/s/submission-guidelines#loc-laboratory-protocols

Laboratory protocols

To enhance the reproducibility of your results, we recommend and encourage you to deposit laboratory protocols in protocols.io, where protocols can be assigned their own persistent digital object identifiers (DOIs).

To include a link to a protocol in your article:

1. Describe your step-by-step protocol on protocols.io

2. Select Get DOI to issue your protocol a persistent digital object identifier (DOI)

3. Include the DOI link in the Methods section of your manuscript using the following format provided by protocols.io: http://dx.doi.org/10.17504/protocols.io.[PROTOCOL DOI]

At this stage, your protocol is only visible to those with the link. This allows editors and reviewers to consult your protocol when evaluating the manuscript. You can make your protocols public at any time by selecting Publish on the protocols.io site. Any referenced protocol(s) will automatically be made public when your article is published. Thank you for your suggestion. We have deposited a short laboratory protocol in protocols.io platform and added the DOI number at the end of Method section, Line 125-127.

 Thank you for your suggestion. We have tried to reorganize all comments and our responses in this table to make better history of publication process.

No. Reviewers' comments Author Responses

1. of a haplotype (typo A) Thank you for your correction. We have fixed it.

2. explain how genetic information can be used to trave back the origin of shark find from this species found in markets, particularly in Asia. Cite Fields et al 2020, in Animal Conservation, a manuscript that came out this week.

 Thank you for your suggestion. We have added introduction in Line 53-67 and cited from Fields et al and Jabado et al in Line 68-75.

 DNA extraction

 Thank you for your correction. We have fixed it.

3. I dont think you need to inlude this, as this is know by population geneticists and also because it will also depend of the probability of significance.

 Thank you for your suggestion. We agreed with your suggestion, we have omitted the sentence.

4. No significant differentiation was observed

(Line 203) Thank you for your correction. We have revised the sentence.

5. significant differentiation Thank you for your correction. We have revised the phrase.

6. differentiation Thank you for your correction. We have revised it.

7. include number of Fst and p value

 Thank you for your suggestion. We have added value of Fst and p in the sentence Line 210.

8. is a highly Thank you for your correction. We have revised the phrase.

9. what type of fisheries, be more specific

 Thank you for your suggestion. We have added more specific information in Line 275.

“according to the global fisheries information system on hammerhead shark capture production 1950-2017 on global scale and Indonesia by FAO”

10. you have many nucleotide changes between your haplotypes from Indonesia and those from the Western Indian Ocean...what could be happening there? is there really gene flow with the West Indian Ocean? I think gene flow will be restricted, otherwise you would not find such differentiated haplotypes. Re -think this section and go deeper into your results

 Thank you for your correction. We have stated some sentences in Line 318-330 regarding haplotype differentiation between two haplogroup between Indonesia and western Indian Ocean, and we also highlight in next sentence 331-347, an indication on genetic sharing between population in Aceh and Western Indian Ocean which have similarity in H1 (haplotype one).

11. and gene flow

 Thank you for your correction. We have added the phrase in Line 339.

---

## [Editor Report · Decision Letter 3]

13 May 2020

PONE-D-20-06557R3

Genetic connectivity of the scalloped hammerhead shark Sphyrna lewini (Griffith & Smith 1834) population across Indonesia and the Western Indian Ocean

PLOS ONE

Dear Dr.Madduppa,

Thank you for submitting your manuscript to PLOS ONE. After careful consideration, we feel that it has merit but does not fully meet PLOS ONE’s publication criteria as it currently stands. Therefore, we invite you to submit a revised version of the manuscript that addresses the points raised during the review process.

There are still issues with the way some of the results, particularly in the abstract, are written.  You use, for example, insignificant, that are not the proper way to refer to your results from a population genetics perspective.  Also, be careful when you make assumptions such as the fact that haplotypes being shared in a network are proof of current gene flow.  This result could also be the result of ancestral sharing of haplotypes.  I am including a PDF with these observations and I ask you to consider them carefully and improve these points.

We would appreciate receiving your revised manuscript by 25th of May 2020. To enhance the reproducibility of your results, we recommend that if applicable you deposit your laboratory protocols in protocols.io, where a protocol can be assigned its own identifier (DOI) such that it can be cited independently in the future. For instructions see: http://journals.plos.org/plosone/s/submission-guidelines#loc-laboratory-protocols

We look forward to receiving your revised manuscript.

Kind regards,

Susana Caballero, PhD

Academic Editor

PLOS ONE

---

## [Author Response · Author response to Decision Letter 3]

25 Jun 2020

4th SUBMISSION RESPONSE

No. Reviewers' comments Author Responses

1. be careful when you explain your findings. You never talk about insignificant differentiation. You say non-significant and siginificantly different. This mistake was already in the previous version. You can look as example sifferentpapers on genetics to see examples on how to write this.

 Thank you for your correction.

We have tried to look other paper as a reference to fix it.

2. in a network you can say haplotype sharing but be careful when you say gene flow...haplotype sharing can be the result of gene flow OR from ancestra sharing of alleles.

 Thank you for your suggestions.

We have detected haplotype sharing between population in Aceh and the Western Indian Ocean, but as you said maybe it is not as a result of gene flow. We have added some sentences to reflected our result in abstract and discussion in Genetic Connectivity section.

3. The combination (Line 65) Thank you for your correction. We have fix it.

4. protocols (plural) (Line 125)

 Thank you for your correction. We have fixed it.

5. what about the probability? was it significant or non-significant?

(Line 214) Thank you for your question.

It was significant, we have written the value of Fst and P.

---

## [Editor Report · Decision Letter 4]

2 Jul 2020

PONE-D-20-06557R4

Genetic connectivity of the scalloped hammerhead shark Sphyrna lewini (Griffith & Smith 1834) population across Indonesia and the Western Indian Ocean

PLOS ONE

Dear Dr. Madduppa,

Thank you for submitting your manuscript to PLOS ONE. After careful consideration, we feel that it has merit but does not fully meet PLOS ONE’s publication criteria as it currently stands. Therefore, we invite you to submit a revised version of the manuscript that addresses the points raised during the review process.

**Your submission still requires substantial editing for English grammar and usage. We ask that you please have the manuscript copyedited by either a native-English speaking colleague or a professional copy-editing service. While you may approach any qualified individual or any professional scientific editing service of your choice, PLOS has partnered with American Journal Experts (AJE) to provide discounted services to PLOS authors. AJE has extensive experience helping authors meet PLOS guidelines and can provide language editing, translation, manuscript formatting, and figure formatting to ensure your manuscript meets our submission guidelines. If the PLOS editorial team finds any language issues in text that AJE has edited, AJE will re-edit the text for free. To take advantage of this special partnership, use the following link:  https://www.aje.com/go/plos/.**

We look forward to receiving your revised manuscript.

Kind regards,

Susana Caballero, PhD

Academic Editor

PLOS ONE

---

## [Author Response · Author response to Decision Letter 4]

10 Sep 2020

Dear Editor and Reviewers,

Thank you very much for your opportunity to improve our manuscript. We have followed the instructions and address all reviewers comments and suggestions. We have also asked professional native speaker to check our manuscript, as this the main comment from last submitted revision. Thanks again and we hope now the manuscript is suitable for publication.

Please kindly see the document entitled "Response to Reviewers" for detail on how addressed questions and suggestions.

---

## [Editor Report · Decision Letter 5]

15 Sep 2020

Genetic connectivity of the scalloped hammerhead shark Sphyrna lewini across Indonesia and the Western Indian Ocean

PONE-D-20-06557R5

Dear Dr. Madduppa

We’re pleased to inform you that your manuscript has been judged scientifically suitable for publication and will be formally accepted for publication once it meets all outstanding technical requirements.

Kind regards,

Susana Caballero, PhD

Academic Editor

PLOS ONE

Additional Editor Comments (optional):

Thank you very much for submitting this revised version and thanks for following the recommendation to have your manuscript professionally proof-read by a service. It reads WAY better!!!! Now it not only presents super interesting results but also they are easier to follow and understand!
---

## [Editor Report · Acceptance letter]

21 Sep 2020

PONE-D-20-06557R5

Genetic connectivity of the scalloped hammerhead shark *Sphyrna*
*lewini* across Indonesia and the Western Indian Ocean

Dear Dr. Madduppa:

I'm pleased to inform you that your manuscript has been deemed suitable for publication in PLOS ONE. Congratulations! Your manuscript is now with our production department.

Kind regards,

on behalf of

Dr. Susana Caballero 

Academic Editor

PLOS ONE